# Exacerbation of Hepatic Damage in Endothelial Aquaporin 1 Transgenic Mice after Experimental Heatstroke

**DOI:** 10.3390/biomedicines12092057

**Published:** 2024-09-10

**Authors:** Kaoru Yanagisawa, Kazuyuki Miyamoto, Yoshihiro Wakayama, Satoru Arata, Keisuke Suzuki, Motoyasu Nakamura, Hiroki Yamaga, Takuro Miyazaki, Kazuho Honda, Kenji Dohi, Hirokazu Ohtaki

**Affiliations:** 1Department of Anatomy, School of Medicine, Showa University, 1-5-8 Hatanodai, Shinagawa-ku, Tokyo 142-8555, Japan; 1997-yanagi@med.showa-u.ac.jp (K.Y.); wakayama@med.showa-u.ac.jp (Y.W.); ks07202251@gmail.com (K.S.); motoyasu@med.showa-u.ac.jp (M.N.); h.yamaga.showa@hotmail.com (H.Y.); kzhonda@med.showa-u.ac.jp (K.H.); 2Department of Emergency, Critical Care and Disaster Medicine, School of Medicine, Showa University, 1-5-8 Hatanodai, Shinagawa-ku, Tokyo 142-8555, Japan; kdop@med.showa-u.ac.jp; 3Wakayama Clinic, 2-3-18 Kanai, Machida, Tokyo 195-0072, Japan; 4Department of Biochemistry, Faculty of Arts and Sciences, Showa University, 4562 Kamiyoshida, Fujiyoshida 403-0005, Japan; arata@pharm.showa-u.ac.jp; 5Center for Biotechnology, Showa University, 1-5-8 Hatanodai, Shinagawa-ku, Tokyo 142-8555, Japan; 6Center for Laboratory Animal Science, Showa University, 1-5-8 Hatanodai, Shinagawa-ku, Tokyo 142-8555, Japan; 7Department of Biochemistry, School of Medicine, Showa University, 1-5-8 Hatanodai, Shinagawa-ku, Tokyo 142-8555, Japan; taku@pharm.showa-u.ac.jp; 8Department of Functional Neurobiology, School of Pharmacy, Tokyo University of Pharmacy and Life Science, 1432-1 Horinouchi, Hachioji, Tokyo 192-0392, Japan

**Keywords:** heatstroke, aquaporin 1, liver, macrophage, mouse

## Abstract

Heatstroke induces fluid loss and electrolyte abnormalities owing to high ambient temperature (AT) and relative humidity (RH). Aquaporin 1 (AQP1) is a key protein for water homeostasis; however, its role in heatstroke remains unclear. This study examines endothelial AQP1 in Tie2-Cre/LNL-AQP1 double transgenic (dTG) mice with upregulated Aqp1 in endothelial cells. For experimental heatstroke, mice were exposed to 41 °C AT and >99% RH. Blood, brain, kidney, and liver samples were collected 24 h later. Blood was analyzed for electrolytes and tissue damage markers, and organs were examined using morphological and immunohistological staining for 3-nitrotyrosine (3-NT), AQP1, and Iba-1. No difference in Aqp1 expression was observed in the whole brain; however, it was detected in dTG mice after capillary deprivation. AQP1 immunostaining revealed immunoreaction in blood vessels. After heat exposure, wild-type and dTG mice showed electrolyte abnormalities compared with non-heatstroke wild-type mice. Hepatic damage markers were significantly higher in dTG mice than in wild-type mice. Hematoxylin–eosin staining and 3-NT immunoreactivity in the liver indicated hepatic damage. The number of Iba-1-positive cells adherent to hepatic vasculature was significantly higher in dTG mice than in wild-type mice. This study is the first to suggest that endothelial AQP1 contributes to hepatic damage after heatstroke.

## 1. Introduction

Exposure to a high ambient temperature (AT) and relative humidity (RH) leads to heatstroke, causing fluid loss and electrolyte abnormalities. This influences water homeostasis in organs and cells, induces multiple organ damage, and often leads to cell death [1]. Global warming has increased the incidence of heatstroke worldwide [2]. In Japan, during the summer of 2018, AT reached record values, resulting in over 90,000 heatstroke-related hospitalizations and 150 deaths. In France, over 70,000 people died from heatstroke in the summer of 2003 [3]. Global warming is expected to continue increasing in the future [4,5]. Therefore, understanding the pathogenesis of heatstroke and developing effective therapeutic strategies is important.

Aquaporins (AQPs) are a family of small integral membrane proteins that facilitate the transport of water and small neutral solutes across various biological membranes and are distributed in diverse organs. In mammals, 13 AQPs have been identified that regulate body fluid production, osmotic regulation, and electrolyte balance [6]. AQP1, the first AQP isoform identified in the erythrocyte membrane, is a water-selective transporting protein [7,8]. AQP1 is strongly expressed in the endothelial cells (ECs) of blood vessels outside the brain [9,10], proximal tubules of the kidney [11], choroid plexus of the brain [10], intestines [12], skeletal muscles [13], inner and intralobular ducts of the pancreas, cholangiocytes of the liver, articular cartilages, and intervertebral discs [6]. However, the role of AQP1 in heat stroke has not yet been elucidated.

We recently generated Tie2-Cre/LNL-AQP1 double transgenic (Tie2-Cre/LNL-AQP1 dTG) mice using the Cre/LoxP system. The upregulation of *Aqp1* in ECs under the control of the Tie2 promoter has also been observed. In this study, we examined the role of endothelial AQP1 in Tie2-Cre/LNL-AQP1 dTG mice with upregulated *Aqp1* on ECs. First, we confirmed the upregulation of endothelial AQP1 in the brain using reverse transcription polymerase chain reaction (PCR) and immunohistochemistry. Subsequently, we subjected dTG mice to experimental heatstroke using high AT and RH to mimic summer conditions in temperate-to-tropical regions, including Japan.

## 2. Materials and Methods

### 2.1. Animals

All animal experimental procedures involving euthanasia were approved by the Institutional Animal Care and Use Committee of Showa University (#09032), following the ARRIVE guidelines.

CAG-LNL(STOP)-AQP1 Tg mice were generated using the Cre/LoxP system, as described previously [14]. The complementary DNA (cDNA) for mouse *Aqp1* (831 bp) was cloned using primers EcoR1-Fw (gggaattcaccATGGCCAGTGAAATCAAGAAGAAGC) and Sac1-Rv (gcaggagctcCTATTTGGGCTTCATCTCCAC). Next, the cDNA was inserted between the EcoRI and Sac1 sites of pCALNL5, Vector Lab, Burlingame, CA USA, and the mouse *Aqp1* DNA was confirmed via sequencing [15]. The fragments were then microinjected into 0.5-day fertilized eggs using a micromanipulator, and the eggs were transferred to the fimbriae of the uterine tubes of female Imprinting Control Region mice that had been mated with vasoligated male mice 1 day prior. The tails (5 mm in length) of 4-week-old pups were cut off, lysed with proteinase K (Wako, Tokyo, Japan), and purified using an automatic nucleic acid isolation system (Kurabo NA-2000, Tokyo, Japan) for genotyping by PCR.

Female founders were mated with male C57BL/6 mice to confirm germline transmission by PCR genotyping, and CAG-LNL(STOP)-AQP1 Tg mice were obtained. CAG-LNL(STOP)-AQP1 Tg mice were then crossed with Tie2-Cre Tg mice (stock #8863, Jackson Laboratory, Bar Harbor, ME, USA), which carried the Cre recombinase transgene in ECs under the control of the Tie2 promoter [16,17]. This step resulted in the removal of the neor cassette from the LNL-CAST transgene, thereby activating the transgene in ECs. Tie2-Cre/LNL-AQP1 dTG mice with a B6 background were designed for *Aqp1* expression in ECs. Wild-type mice were obtained from their littermates.

### 2.2. Collection of Capillary-Rich Fraction (CF) in the Brain and mRNA Isolation

To confirm the dTG mice, we checked the gene expression of *Aqp1* in the brain. Brain ECs of choroid plexus express AQP1, but ECs of capillaries generally do not express *Aqp1* [9]. Moreover, the primary cells of ECs in rat capillaries express low-level AQP1 [18].

CF was collected as previously described, with minor modifications [19,20]. After cervical dislocation in the male the telencephalon was freshly obtained, and the pia mater was carefully removed with a paper towel. The brain was then homogenized with 4.2 mL of capillary buffer (10 mM HEPES, 141 mM NaCl, 4 mM KCl, 2.8 mM CaCl_2_, 1 mM NaH_2_PO_4_, 1 mM MgSO_4_, 10 mM D-glucose, pH 7.4) using a loose glass Dounce tissue grinder (10 strokes, Wheaton, Millville, NJ, USA). Then, 10.2 mL of 26% dextran (Sigma, St Louis, MO, USA) was added and gently mixed. The homogenate was centrifuged at 4300× *g* in a swing-bucket rotor for 30 min, and the supernatant was removed. The pellet was resuspended in 10 mL of capillary buffer and centrifuged at 8000× *g* for 30 min. The pellet was then mixed with TRIzol Reagent (Invitrogen, Carlsbad, CA, USA), and mRNA was isolated [21]. The purity and concentration of extracted RNA were determined using a spectrophotometer (NanoDrop, Wilmington, DE, USA). A High-Capacity RNA-to-cDNA Kit (Applied Biosystems, Foster City, CA, USA) was used to synthesize cDNA using 2 μg total RNA, according to the manufacturer’s instructions. The experiment was repeated twice to assess reproducibility.

### 2.3. PCR

PCR was performed using TaKaRa Ex Taq (TaKaRa Bio, Shiga, Japan). The reaction mixture was prepared with an appropriate volume of cDNA mixture containing the following: 0.25 μL of forward and reverse primers (50 nmol/mL), 2.0 μL of deoxyribonucleotide (dNTP) mixture (0.25 mM each), 0.1 μL of TaKaRa Ex Taq (5 units/μL), and 2.0 μL of 10× Ex Taq Buffer in a total volume of 20 μL. Thermal cycling parameters were set as follows: 95 °C for 1 min for initial denaturation, 30–40 cycles at 95 °C for 45 s, 55 °C or 60 °C for 30 s, and 72 °C for 45 s. At the end of the final cycle, an additional 7 min extension step was performed at 72 °C. Ten microliters of each reaction mixture were electrophoresed on a 2.0% agarose gel, and the bands were visualized using ethidium bromide. Mouse ribosomal protein S18 (*Rps18*) served as the housekeeping gene. Primers used are listed in Table 1.

### 2.4. Multiple Staining for AQP1 Localization

To visualize AQP1 localization on ECs in the brain, we labeled ECs with tomato lectin and multiple immunostaining techniques in situ. Following an overdose of sodium pentobarbital (100 mg/kg, i.p.) anesthesia, the male (n = 3) aged 11 months were transcardially perfused with 0.9% NaCl to remove blood. Subsequently, 1% paraformaldehyde was injected. The mice were then perfused with 10 mL of 5 μg/mL DyLight 488-labeled Lycopersicon Esculentum Tomato lectin (Vector Lab, Burlingame, CA, USA), and the brains were collected. The brains were immersed in 20% sucrose in 0.1 M phosphate buffer (pH 7.2; PB) for two nights and then embedded in liquid nitrogen-cooled isopentane using an embedding solution (20% sucrose in 0.1 M PB: O.C.T. compound (Sakura Finetech, Tokyo, Japan, 2:1)). Coronal sections (8 μm thick) were obtained using a cryostat (Hyrax50, Carl Zeiss, Oberkochen, Germany), and they were immunostained.

After washing with phosphate-buffered saline (PBS), sections were subjected to heat-mediated antigen retrieval using 10 mM sodium citrate (pH 6.0) at 95 °C for 25 min. The sections were washed with PBS containing 0.05% Tween 20 and incubated with MOM mouse IgG blocking reagent (Vector Lab, Burlingame, CA, USA). The sections were blocked with 5% normal horse serum (NHS) (Vector Lab, Burlingame, CA USA) for 60 min and subsequently incubated with mouse anti-rat AQP1 (1/22) (1:200, Santa Cruz Biotechnology, Santa Cruz, CA, USA, cat# sc-32737) and rabbit anti-GFAP (1:10, DAKO, Glostrup, Denmark, cat# N1506) antibodies at 4 °C overnight. The sections were then incubated with the appropriate Alexa-labeled goat anti-mouse (1:400, Invitrogen, Carlsbad, CA, USA, cat# A11029 and A11030) and goat anti-rabbit (1:400, Invitrogen, Carlsbad, CA, USA, cat# A21068) secondary antibodies at 25 °C or 2 h. In some cases, the cell nuclei were stained with 4,6-diamidine-2-phenylindole dihydrochloride (1:10,000; Roche, Mannheim, Germany) and then incubated in 1.0 mM CuSO_4_ in 50 mM ammonium acetate buffer (pH 5.0) to reduce autofluorescence [22]. Fluorescence was detected using an Axio Imager optical sectioning microscope equipped with ApoTome (Carl Zeiss, Baden-Wurttemberg, Germany).

### 2.5. Heatstroke Model

A semi-enclosed acrylic heatstroke chamber (200 mm × 340 mm × 300 mm) was custom-made by placing one animal cage on top of another, similar to a greenhouse [23]. An ultrasonic humidifier (USB-68, Sanwa, Okayama, Japan) and a digital thermo-hygrometer (AD-5696, A&D Company, Tokyo, Japan) were used for the humidification and monitoring of AT, RH, and wet-bulb globe temperature (WBGT). The WBGT is an environmental index that accounts for AT and RT in a room [24,25] and is recommended by the International Labor Organization for evaluating workers’ environments. The heatstroke chamber was placed in an incubator (Bio-chamber, BCP-120F, TITEC, Aichi, Japan), which was pre-heated at 41 °C AT and >99% RH for 3 h or more.

Eleven-month-old male mice were weighed and dehydrated 3 h prior to heat exposure. Following this, the mice were placed in a heatstroke chamber for 60 min and then returned to room temperature (23 ± 1 °C) with free access to food and water according to previous studies [26,27,28]. Body weight (BW) was measured four times as follows: before dehydration, immediately before and after heat exposure, and after 24 h of heat exposure. Nine mice (four wild-type and five Tie2-Cre/LNL-AQP1 dTG mice) were subjected to heat exposure.

After 24 h of heat exposure for 1 h, the mice were anesthetized with an overdose of sodium pentobarbital (100 mg/kg, i.p.). Blood samples were collected from the hearts’ right ventricles, and the mice were transcardially perfused with 0.9% NaCl and 10% neutralized formalin. Blood samples were centrifuged at 1500× *g* for 10 min to collect serum. The serum samples were examined for biochemical parameters. Fixed organs, including the liver, kidney, and brain, were collected, paraffin-embedded 4 μm thick sections were prepared, and morpho-histological changes were evaluated.

### 2.6. Serum Biochemical Parameters

Serum levels of total protein (TP), albumin (ALB), total bilirubin (T-bil), aspartate aminotransferase (AST), alanine aminotransferase (ALT), alkaline phosphatase (ALP), lactate dehydrogenase (LDH), creatinine kinase (CK), blood urea nitrogen (BUN), creatinine (Cre), electrolytes (Na^+^, K^+^, and Cl^−^), glucose, lactate (LA), and osmotic pressure were analyzed. TP was analyzed using the biuret method; ALB was analyzed using the BCG method; BUN was analyzed using the urease–GLDH method; T-bil, Cre, and LA were analyzed using the enzymatic method; AST, ALT, ALP, LDH, and CK were analyzed using the MDH-UV method; electrolytes were analyzed using the ion-selective electrode method; and osmotic pressure was analyzed using the freezing point depression method. All assays were performed using a HITACHI 7180 (Hitachi, Tokyo, Japan).

### 2.7. Immunohistochemistry and Cell Counts

Paraffin-embedded sections were deparaffinized using a series of concentrations of xylene and alcohol and then subjected to hematoxylin–eosin and immunohistochemical staining.

After washing with PBS, the sections were subjected to heat-mediated antigen retrieval with 10 mM sodium citrate buffer (pH 6.0) at 95 °C for 25 min. Subsequently, they were immersed in 0.3% H_2_O_2_/MeOH for 30 min. For AQP1 immunostaining, the sections were blocked with MOM mouse IgG. They were further blocked with 5% NHS to prevent non-specific binding. They were incubated with rabbit anti-Iba1 (1:500, Wako, Osaka, Japan, cat# 019-19741), rabbit anti-3-nitrotyrosine (1:500, Upstate Biotechnology, Lake Placid, NY, USA cat# 06-284), and mouse anti-rat AQP1 (1/22) (1:200) antibodies at 4 °C overnight. The sections were then incubated with biotinylated goat anti-rabbit (Invitrogen, Carlsbad, CA, USA cat# A16114) or goat anti-mouse IgG (DAKO, Glostrup, Denmark, cat# E0433) secondary antibodies for 2 h. They were subsequently incubated in an avidin–biotin complex solution (Vector Lab, Burlingame, CA, USA) and diaminobenzidine (Sigma, Milwaukee, WI, USA) as a chromogen. Sections were observed under a microscope (Olympus BX53; Olympus, Tokyo, Japan), and images were captured with an Olympus cellSens standard 2.1 (Olympuonvs, Tokyo, Japan). Staining was conducted on sections obtained from four and five wild-type and Tie2-Cre/LNL-AQP1 dTG mice, respectively.

For multiple staining of Iba1 and AQP1, the sections were co-incubated with rabbit anti-Iba1 and mouse anti-rat AQP1 antibodies and visualized with Alexa 546 goat anti-mouse and Alexa 488 goat anti-rabbit (1:400, Invitrogen, Carlsbad, CA, USA, cat# A11034) antibodies.

To determine the influence of AQP1 on monocyte and macrophage infiltration, we manually counted the number of monocytes and macrophages attached to the vasculature after Iba1-immunostaining in the brain, liver, and kidney, using the process mode in Olympus cellSens standard 2.1. The number of Iba1+ immunoreactions (irs) was counted in 30 vessels of the telencephalons and livers and 20–24 vessels in the kidneys from 3–4 sections in each mouse. The boundary lengths of the counted vessels were measured, and the estimated diameter of each vessel was calculated. Cells were counted in small- and medium-sized vessels, excluding capillaries and sinusoids of the liver. An investigator (M.N. and H.Y.) blinded to the mouse genotype performed the counting.

### 2.8. Statistical Analysis

Data are expressed as mean ± standard error of the mean. Student’s *t*-test was used for comparisons between two groups. One-way analysis of variance and Tukey’s post hoc test were used to perform multiple comparisons. A value of *p* < 0.05 was considered statistically significant. Analyses were performed using the Bell Curve for Excel (Bell Curve, Tokyo, Japan).

## 3. Results

### 3.1. Evaluation of Tie2-Cre/LNL-AQP1 dTG Mice

AQP1 was first identified in erythrocyte membranes [29] and is expressed in the kidneys, lungs, heart, liver, cornea, blood vessels, and choroid plexus [6]. However, it is not usually expressed in the blood vessels of the brain [9]. Therefore, we examined *Aqp1* expression in the capillaries of the telencephalon. We evaluated the specificity of Tie2-Cre/LNL-AQP1 dTG mice. *Aqp1* expression was similarly detected in whole-brain homogenates of both wild-type and Tie2-Cre/LNL-AQP1 dTG mice (Figure 1A). We collected CFs to compare *Aqp1* expression in capillary ECs. CFs were validated using transcript levels of ECs and other neural cell markers. Since the vasculature transcript levels of *Pecam* (with two sets of primers) and *Vwf* were similarly expressed in both types of animals, the brains of Tie2-Cre/LNL-AQP1 dTG mice suggest an increase in *Aqp1* expression. Moreover, CFs showed lower expression of neuronal or glial transcripts such as *Eno2*, *Gfap*, *Mbp*, and *Aif1* than the whole brain fraction. This also indicated that the brains of Tie2-Cre/LNL-AQP1 dTG mice showed increased *Aqp1* expression in the vasculature (Figure 1A).

We then observed the stained brain sections for the AQP1 antibody (Figure 1B,C). AQP1-ir was observed in the neuronal layer of the piriform cortex in wild-type mice. However, AQP1-ir in Tie2-Cre/LNL-AQP1 dTG mice was observed in both vasculature-like structures and in the neuronal layer of the cortex. Next, we examined the in situ vasculature with tomato lectin and performed multiple staining for AQP1. Although AQP1-ir in wild-type mice was weakly observed in the same field at higher magnification, AQP1-ir did not co-localize with tomato lectin signals and was mostly co-localized with GFAP-ir. In contrast, in Tie2-Cre/LNL-AQP1 dTG mice, AQP1-ir co-localized with tomato lectin signals, indicating its expression in ECs (Figure 1C).

### 3.2. Changes in AT, RH, and WBGT during Heat Exposure

The animals were exposed to experimental heatstroke and subjected to high AT and RH for 60 min. Figure 2A shows the trends in AT, RH, and WBGT in the semi-enclosed heatstroke chamber. During the 3 h incubation and before heat exposure, AT and RH values inside the chamber were 38.1 °C and >99.9%, respectively, and WBGT was calculated at 42.0 °C. Although these conditions temporarily decreased when the animals were dropped into the chamber, AT and WBGT gradually increased and reached 39.9 °C and 43.9 °C after 60 min. The animals exhibited jumping behavior approximately 40 min after the exposure and gathered at the corner of the chamber. However, no animals died within 60 min of heat exposure or 1 d after recovery. The changes in BW during the experimental period are shown in Figure 2B. Compared with prior hydration, BW decreased to approximately 96.5% after heat exposure and returned to baseline levels 1 d after heat exposure. No significant differences were observed between the groups.

### 3.3. Serum Biochemical Parameters after 1 Day of Heat Exposure

Electrolytes, including Na^+^, K^+^, and Cl^−^, and osmotic pressures are shown in Figure 3. After heat exposure, Na^+^ and K^+^ decreased, while Cl^−^ increased in both experimental groups compared with those in non-heat-exposed wild-type animals. Compared with non-heat-exposed wild-type animals, significant differences were observed for all electrolytes in wild-type mice and only for Cl^−^ in Tie2-Cre/LNL-AQP1 dTG mice after heat exposure.

Next, we assessed the levels of tissue damage markers in the serum after heat exposure (Figure 4). Most markers did not show significant differences between wild-type and Tie2-Cre/LNL-AQP1 dTG mice after heat exposure. However, the level of LDH, a tissue damage marker, was significantly higher in Tie2-Cre/LNL-AQP1 dTG mice than in wild-type mice (*p* < 0.05). Moreover, the hepatic damage markers ALT and AST were increased in Tie2-Cre/LNL-AQP1 dTG mice; however, the difference was not significant. These results suggest that Tie2-Cre/LNL-AQP1 dTG mice may experience increased tissue damage, likely hepatic damage, compared with wild-type mice after heat exposure.

### 3.4. Expression of Hepatic AQP1 on Vessels

We determined the localization of AQP1 in the liver at 24 h after heat exposure (Figure 5). AQP1-ir was ubiquitously expressed in the hepatic vasculature, including the sinusoids of non-heat-exposed wild-type mice. It showed no staining in the portal triad’s hepatocytes and small bile ducts. There was no remarkable difference in AQP1-ir staining between the wild-type and Tie2-Cre/LNL-AQP1 dTG mice after heat exposure.

### 3.5. Increase in Inflammation in the Liver of Tie2-Cre/LNL-AQP1 dTG Mice

After heat exposure, we compared the hepatic morphological alterations between wild-type and Tie2-Cre/LNL-AQP1 dTG mice (Figure 6). Hepatocytes in non-heat-exposed wild-type mice exhibited large, round nuclei with distinct outlines. The sinusoid radiates outward from the central vein. After heat exposure, the nuclei in the hepatocytes of wild-type mice shrank and were deeply stained. Sinusoids were still radially organized, and few leukocytes were observed. After heat exposure, the hepatic lobules in Tie2-Cre/LNL-AQP1 dTG mice exhibited a disorganized sinusoidal arrangement, and many leukocytes were observed. The nuclei of the hepatocytes shrank and became deeply stained. Morphological observations suggested that the hepatic tissues of Tie2-Cre/LNL-AQP1 dTG mice were inflamed.

### 3.6. Increase in Oxidative Stress in Tie2-Cre/LNL-AQP1 dTG mice

To confirm hepatic damage, a protein oxidative metabolite, 3-NT (3-nitrotyrosine), was immunostained in the liver (Figure 7). Minimal or no staining was observed in the livers of Tie2-Cre/LNL-AQP 1 dTG mice in the primary antibody-free negative control. Slight 3-NT-ir was detected in wild-type mice and was prominent in the vessels, including the sinusoids, of the livers in Tie2-Cre/LNL-AQP1 dTG mice. This result further supports the increased hepatic damage observed in Tie2-Cre/LNL-AQP1 dTG mice.

### 3.7. Increase in Iba1-Positive Cells on Vessels in the Liver of Tie2-Cre/LNL-AQP1 dTG Mice

Next, we stained hepatic tissues with an anti-Iba1 antibody (Figure 8). Iba1 is a marker of monocytes and macrophages. Iba1-ir did not show a remarkable difference in the sinusoids between wild-type and Tie2-Cre/LNL-AQP1 dTG mice, suggesting the presence of Kupffer cells (Figure 8A,B). However, Iba1-ir was highly noticeable in the small- and medium-sized vessels of Tie2-Cre/LNL-AQP1 dTG mice (Figure 8C–F). The tie2 gene has been reported to be expressed in monocyte and macrophage lineage cells [30,31]. Therefore, we hypothesized that Tie2-Cre/LNL-AQP1 dTG mice would show increased expression of AQP1 in Iba1^+^ monocytes/macrophages and could play an important role in the inflammatory response without contributing to endothelial AQP1 after heat exposure. The livers of dTG mice were stained with Iba1 and AQP1 antibodies (Figure 8G).

Although both AQP1-ir and Iba1-ir were closely observed, they did not co-stain. Similar results confirmed that Iba1-ir did not co-localize with AQP1-ir in the kidneys, spleen, or intestines of Tie2-Cre/LNL-AQP1 dTG mice (Figure 9).

Next, the vasculature adhering to Iba1^+^ cells in the liver was counted among the groups. Moreover, to compare the livers, Iba1^+^ cells were counted in the brain’s telencephalic vessels and the kidneys’ interlobular vessels (Figure 9). Generally, the brain does not express AQP1 in wild-type mice, as shown in Figure 1, and the kidneys are critical for water homeostasis. Iba1^+^ numbers on the hepatic vessels of wild-type mice with and without heat exposure were 1.76 ± 0.16 and 2.55 ± 0.53/mm, respectively. The Iba1^+^ number on the hepatic vessels of Tie2-Cre/LNL-AQP1 dTG mice was 5.36 ± 1.45 (*p* < 0.01), and it was significantly greater than that of the wild-type mice both with and without heat exposure (Figure 10A). In contrast, the numbers in the brain and kidneys were less than 2.2/mm in all groups and were not significantly different (Figure 10B,C).

The relationship between Iba1^+^ cells and vessel diameter was assessed in wild-type and Tie2-Cre/LNL-AQP1 dTG mice (Figure 10D) to determine the role of AQP1 in hepatic vessels. While the number of Iba1^+^ cells showed a higher trend in all sizes of vessels in Tie2-Cre/LNL-AQP1 dTG mice than in wild-type mice, the numbers were significantly greater in Tie2-Cre/LNL-AQP1 dTG mice, particularly in relatively small vessels (0–150 and 150–300 μm), compared with heat-exposed wild-type mice.

## 4. Discussion

AQP1 is a water-selective transport protein that plays an important role in fluid production, osmotic regulation, and electrolyte balance [6]. Heatstroke causes body fluid loss and electrolyte abnormalities due to exposure to high AT and RH [1]. Appropriate rehydration to maintain osmotic and electrolyte balance is an important strategy for decreasing the risk of heatstroke [23]. We previously established an experimental heat stroke model in mice and found that the mice impaired livers, kidneys, intestines, and cerebellum in the brain depending on severity [27,28]. However, the role of AQP1 in heat stroke has not yet been determined. In the present study, we generated Tie2-Cre/LNL-AQP1 dTG mice using the Cre/LoxP system to upregulate *Aqp1* in ECs under the control of the Tie2 promoter. The mice were then subjected to experimental heatstroke by exposure to high AT and RH to evaluate the role of vascular AQP1. To confirm the genetic modification in Tie2-Cre/LNL-AQP1 dTG mice, we first assessed AQP1 levels in the telencephalon because ECs in the brain do not normally express AQP1 [9].

*Aqp1* expression in the whole brains of both wild-type and Tie2-Cre/LNL-AQP1 dTG mice was similar. However, after CF collection, *Aqp1* expression was detected only in Tie2-Cre/LNL-AQP1 dTG mice. In contrast, the endothelial marker genes *Pecam* and *Vwf* levels were similar in both wild-type and Tie2-Cre/LNL-AQP1 dTG mice. AQP1 immunostaining supported these results. In the brain, AQP1 is strongly localized to the apical membrane of the choroid plexus [32], neurons, and astrocytes [6,33]. Immunostaining revealed that the neuronal layers of the piriform cortex and astrocytes, which showed very low intensities, were also AQP1-ir. The brains of Tie2-Cre/LNL-AQP1 dTG mice showed clearly labeled vasculature-like structures that co-localized with ECs.

In the present study, the mice were initially exposed to AT and RH of 38.1 °C and >99.9%, respectively, and after 60 min, AT and RH became 39.9 °C and >99.9%, respectively. WBGT was calculated at 42.0 °C and 43.9 °C. A WBGT value above 31 °C in Japan is considered dangerous for human health. Therefore, heat stress is considered to be extremely severe but sublethal. After heat exposure, wild-type and Tie2-Cre/LNL-AQP1 dTG mice showed a 3–4% decrease in BW, with no significant difference. Serum Na^+^ and K^+^ levels decreased, and Cl^−^ levels increased in Tie2-Cre/LNL-AQP1 dTG mice compared with those in non-heat-exposed wild-type mice. After heat exposure, no difference was observed between wild-type and Tie2-Cre/LNL-AQP1 dTG mice. Although the osmotic pressures differed after heat exposure owing to large deviations, heat-exposed animals showed early symptoms of heatstroke [1].

We compared serum biochemical parameters and observed that the cellular damage marker LDH was significantly higher in Tie2-Cre/LNL-AQP1 dTG mice than in wild-type mice. Moreover, the hepatic damage markers AST and ALT also increased in Tie2-Cre/LNL-AQP1 dTG mice; however, the increase was not significant. Therefore, we further analyzed hepatic morphology. Hepatic morphology and oxidative stress detected by 3-NT-ir also suggested hepatic impairment in Tie2-Cre/LNL-AQP1 dTG mice.

A few studies have reported the expression of AQPs in heatstroke and related conditions but not in the liver. *Aqp1* expression increased in the intestines of anesthetized rats after 1 h of heat exposure and returned to baseline level within 12 h [12]. Clinical case reports of sauna-associated death [34] and fatal heat stroke [35] have shown increased AQP3-ir in the skin epidermis and *Aqp4* expression in the brain. However, no studies have suggested a role for AQP1 in hepatic damage after heatstroke, although a role for AQP1 in hepatic damage has been suggested in hepatitis models. AQP1 expression increases in cirrhotic liver ECs following bile duct ligation. AQP1 gene-deficient (KO) mice with cirrhosis show decreased angiogenesis, fibrosis, and portal hypertension, which depend on osmotically sensitive microRNAs in the AQP1 pathway [36,37,38]. These reports including our study suggest that either increased AQP1 expression in ECs under stresses exacerbates hepatic injury or a signal of hepatic damage induced the expression and/or translocation of AQP1. A caspase-3 inhibitor prevented pulmonary injury induced by common bile duct ligation, an experimental model of hepatopulmonary syndrome, and decreased apoptosis and endothelial AQP1 levels [39]. Fibroblasts obtained from gene-deficient mice with N-glycanase 1 and human gene-silenced fibroblasts showed decreased *Aqp*1 levels and were resistant to hypotonic lysis [40].

In contrast, some studies have shown that AQP1 induction decreases inflammation and apoptosis in hepatic and pulmonary injuries. Hepatocyte-induced AQP1 using adenoviruses improves estrogen-induced and lipopolysaccharide-induced cholestasis in rats [41,42,43]. Alpinetin, a Chinese medicine, reduces lung AQP1 levels [44]. In vitro, the induction of the flow-responsive transcription factor, Krüppel-like factor 2 (KLF2), is accompanied by the induction of AQP1. KLF2 maintains an anticoagulant and anti-inflammatory endothelium with sufficient nitric oxide bioavailability. Thus, endothelial expression of AQP1 is associated with an atheroprotected, non-inflamed vessel wall [45].

In the present study, we generated mice with high AQP1 expression in ECs. AQP1-ir in the liver was observed in the intralobular vessels and sinusoids but not in the portal triad’s hepatocytes or intralobular bile ducts. Although we demonstrated that AQP1 expression in ECs increased in the brains of Tie2-Cre/LNL-AQP1 dTG mice, it is still uncertain that it is increased in the hepatic vasculature of Tie2-Cre/LNL-AQP1 dTG mice because the native hepatic vasculature highly expresses AQP1. The role of AQP1 in human patients with heatstroke requires further investigation because the localization of AQP1 in the liver can vary among species. The localization of AQP1 in rodents has mainly been reported in ECs, but the levels are minimal or low in bile ducts and hepatocytes [36,37,38], similar to our results. However, AQP1 localization in pigs and humans is not only observed in the blood vessels but also in the bile ducts and hepatocytes [46,47], and AQP1 is more dominant in bile ducts than in blood vessels [48,49].

Finally, we observed that vasculature-adherent Iba-1^+^ monocytes and macrophages increased in the small- and medium-sized intralobular vessels of Tie2-Cre/LNL-AQP1 dTG mice but were not prominent in sinusoidal Iba-1^+^ cells or Kupffer cells. The tie2 gene is known to be expressed in monocyte/macrophage lineage cells [30,31]. Therefore, we examined multiple staining of Iba1 and AQP1 to eliminate the possibility that Tie2-Cre/LNL-AQP1 dTG mice had increased AQP1 expression in monocytes/macrophages and the AQP1-expressing monocyte/macrophage directly contributed to hepatic damage without mediating endothelial AQP1 expression. Histologically, we did not observe co-localization of AQP1 with monocytes/macrophages. These findings were also observed in the resident macrophages of other organs.

No increase in the number of Iba-1^+^ cells was observed in the telencephalic or renal vessels of Tie2-Cre/LNL-AQP 1 dTG mice. These results suggest that increased hepatic damage in Tie2-Cre/LNL-AQP1 dTG mice induces monocyte and macrophage migration. In particular, Iba-1^+^ monocytes and macrophages in Tie2-Cre/LNL-AQP1 dTG mice were significantly greater in small vessels, likely because of impaired hepatic circulation, because the sinusoids in Tie2-Cre/LNL-AQP1 dTG mice were unclear and partially obstructed.

Together, these results suggest that an increase in endothelial AQP1 may contribute to hepatic injury, depending on electrolyte abnormalities and osmotic changes. The expression of endothelial AQP1 may also correlate with the induction of endothelial adhesion molecules and the acquisition of circulating leukocytes. Myocardin-related transcription factor-A KO mice showed significantly attenuated neointimal formation and decreased expression of *Aqp1*. Moreover, *Icam1*, *Mmp9*, and *Itgb1* expressions decrease in KO mice [50]. Single nucleotide polymorphisms in the priapism in patients with sickle cell disease are associated with TGFBR3, AQP1, and integrin-αv [51]. Further studies are required to clarify the properties of hepatic vessels and AQP1 after heatstroke.

## 5. Conclusions

In the present study, we generated Tie2-Cre/LNL-AQP1 dTG mice with upregulated *Aqp1* expression in ECs under the control of the Tie2 promoter. After severe heat exposure, although both the wild-type and Tie2-Cre/LNL-AQP1 dTG mice exhibited electrolyte abnormalities, Tie2-Cre/LNL-AQP1 dTG mice showed a significant increase in liver injury, 3-NT levels, and hepatic vasculature-adherent monocytes/macrophages. These results suggest that endothelial AQP1 contributes to hepatic damage following heatstroke. These findings indicate that targeting endothelial AQP1 could prevent or mitigate hepatic damage in patients suffering from heatstroke. Future therapeutic approaches could focus on modulating AQP1 expression or activity to protect the liver from heat-induced injury.

## Figures and Tables

**Figure 1 biomedicines-12-02057-f001:**
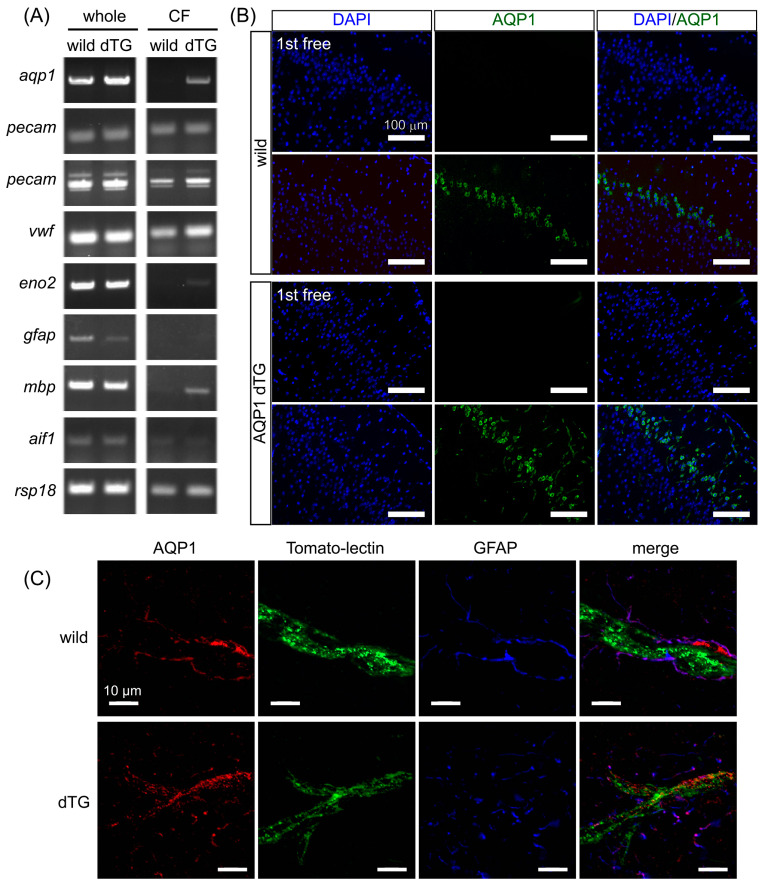
AQP1 expression increases in the endothelial cells (ECs) of Tie2-Cre/LNL-AQP1 dTG mice. AQP1 levels were assessed in the telencephalon of the brain to evaluate whether Tie2-Cre/LNL-AQP1 double transgenic (dTG) mice showed increased AQP1 expression in ECs under the control of the Tie2 promoter. (**A**) *Aqp1* expression was ubiquitously detected in the whole brains of wild-type and Tie2-Cre/LNL-AQP1 dTG mice, but the difference was not significant. After collection of the capillary-rich fraction (CF), the expression levels of endothelial marker genes (two different primer sets of *Pecam* and *Vwf*) and the housekeeping gene (*Rsp18*) were similar in both wild-type and dTG mice. However, the expression of *Aqp1* was greater in dTG mice than in wild-type mice. Whole-brain fractions were also detected in neuronal (*Eno2*), astroglial (*Gfap*), oligodendroglial (*Mpb*), and microglial (*Iba1*) sections but were decreased or not detected in CF, suggesting that CF is rich in ECs. (**B**) Immunofluorescence staining for AQP1 in the brain demonstrated that AQP1 immunoreactivity in the dTG was detected in the vessels and neurons of the piriform cortex. (**C**) Multiple staining of AQP1, endothelial (tomato lectin), and astroglial (GFAP) markers was used to determine whether AQP1 immunoreactivity co-localized with tomato lectin in the dTG.

**Figure 2 biomedicines-12-02057-f002:**
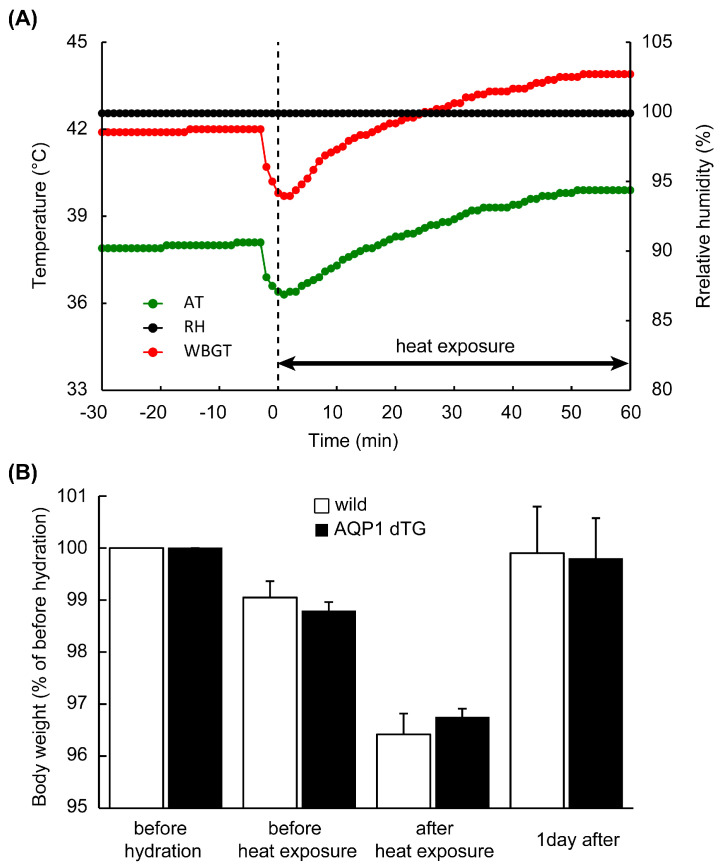
Condition of the heat stroke model and body weight (BW) during the experiment. (**A**) Transition of ambient temperature (AT), relative humidity (RH), and wet bulb globe temperature (WBGT) before and during heat exposure. AT and RH inside the chamber were 38.1 °C and >99.9%, and WBGT was calculated at 42 °C. AT and WBGT temporarily decreased when the animals were placed in the chamber, and after 60 min, AT and WBGT gradually increased, reaching 39.9 and 43.9 °C, respectively. (**B**) Compared with the BWs before hydration, which was 100%, after 1 h of heat exposure, they decreased and mostly returned to baseline levels after 1 day. No significant difference was observed in BW during the experimental periods.

**Figure 3 biomedicines-12-02057-f003:**
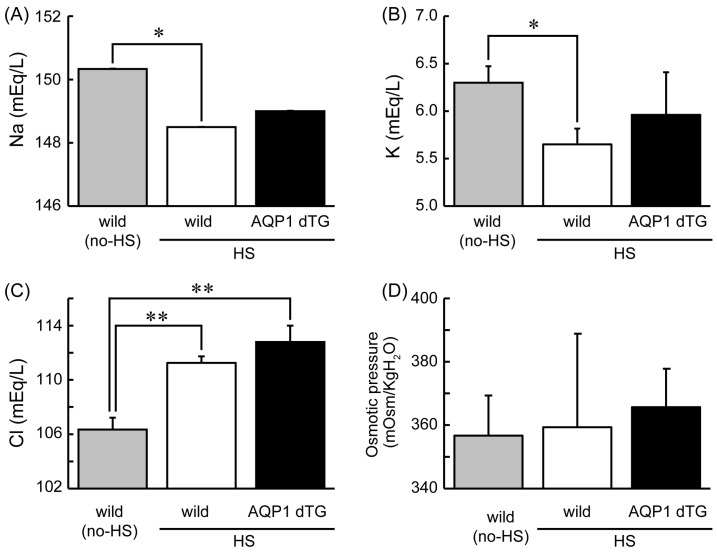
Serum electrolytes and osmotic pressures 1 day after heat exposure. Levels of Na^+^ (**A**), K^+^ (**B**), Cl^−^ (**C**), and osmotic pressure (**D**) in wild-type (wild; *n* = 4) and Tie2-Cre/LNL-AQP1 dTG (AQP1 dTG; *n* = 5) mice. Na^+^ and K^+^ decreased, and Cl^−^ increased in both heat-exposed groups compared with those in non-heat-exposed (no-HS) wild-type animals (*n* = 3). Data are expressed as the mean ± SE. ** p* < 0.05, *** p* < 0.01 (Tukey’s post hoc test).

**Figure 4 biomedicines-12-02057-f004:**
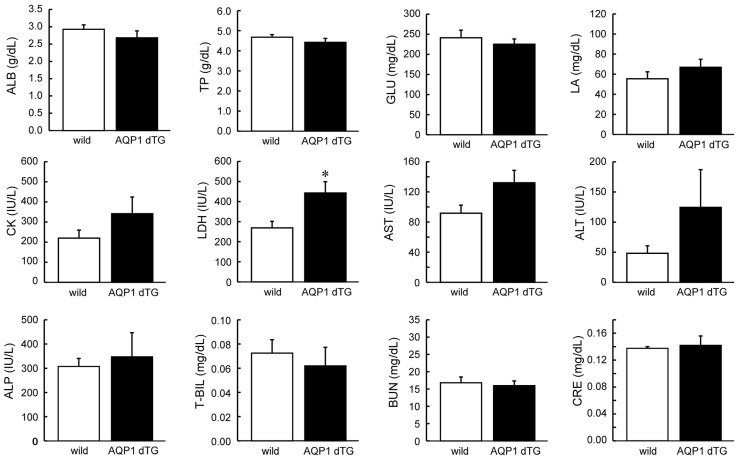
Comparison of serum biochemical parameters, including tissue damage markers. Serum biochemical parameters were determined after 1 day of heat exposure. Lactate dehydrogenase (LDH), a tissue damage marker, was significantly higher in Tie2-Cre/LNL-AQP1 dTG mice (n = 5) than in wild-type mice (n = 4; *p* < 0.05). Data are expressed as the mean ± SE. * *p* < 0.05 (Student’s *t*-test).

**Figure 5 biomedicines-12-02057-f005:**
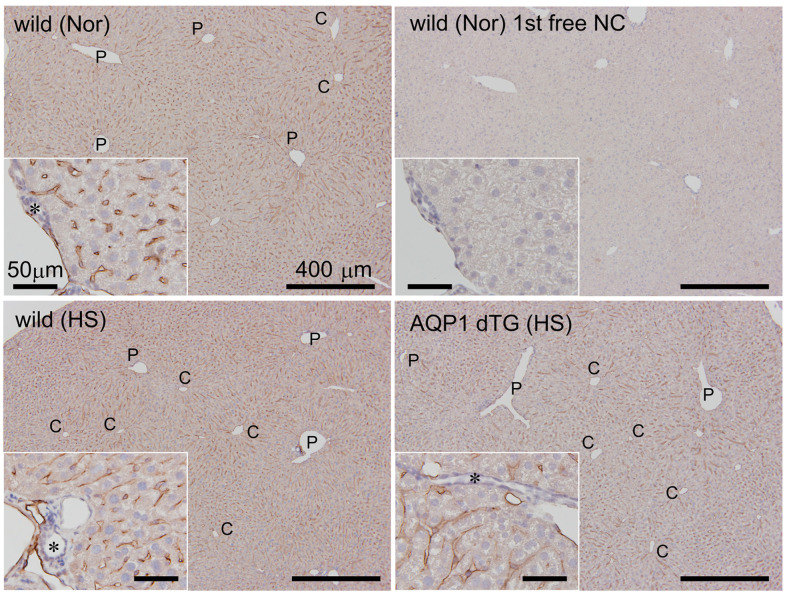
AQP1 immunostaining in the liver. AQP1 immunostaining was performed in the livers of wild-type mice with and without heat exposure and in Tie2-Cre/LNL-AQP1 dTG mice. AQP1-immunoreactions (irs) were ubiquitously observed in the hepatic vasculature, including the sinusoids of non-heat-exposed wild-type mice (Nor). It showed no staining in the portal triad’s (inset) hepatocytes and small bile ducts (asterisk). Minimal or no staining was observed in the primary antibody-free negative control (1st AB-free NC) in the livers of wild-type mice (Nor). There were no remarkable differences in AQP1-ir staining between wild-type (HS) and Tie2-Cre/LNL-AQP1 dTG (HS) mice after heat exposure. C: central vein of the hepatic lobules; P: portal triad.

**Figure 6 biomedicines-12-02057-f006:**
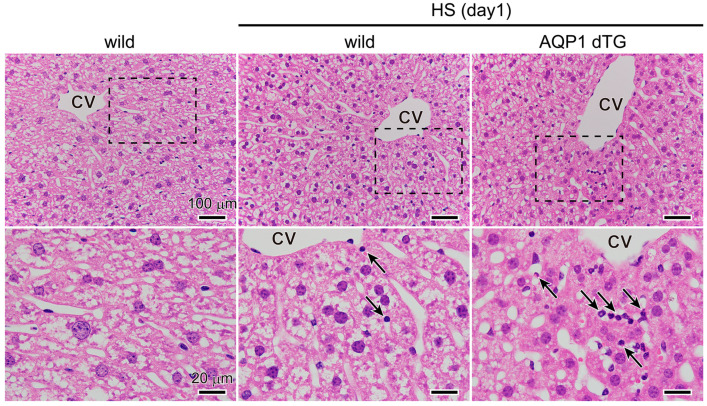
Morphological comparisons of the liver. Representative images of livers after hematoxylin–eosin staining in non-heat-exposed and heat-exposed wild-type and Tie2-Cre/LNL-AQP1 dTG mice. High-magnification images of each rectangle are shown at the bottom. Leukocytes were observed after heat exposure (indicated by arrows). CV: central vein of hepatic lobules.

**Figure 7 biomedicines-12-02057-f007:**
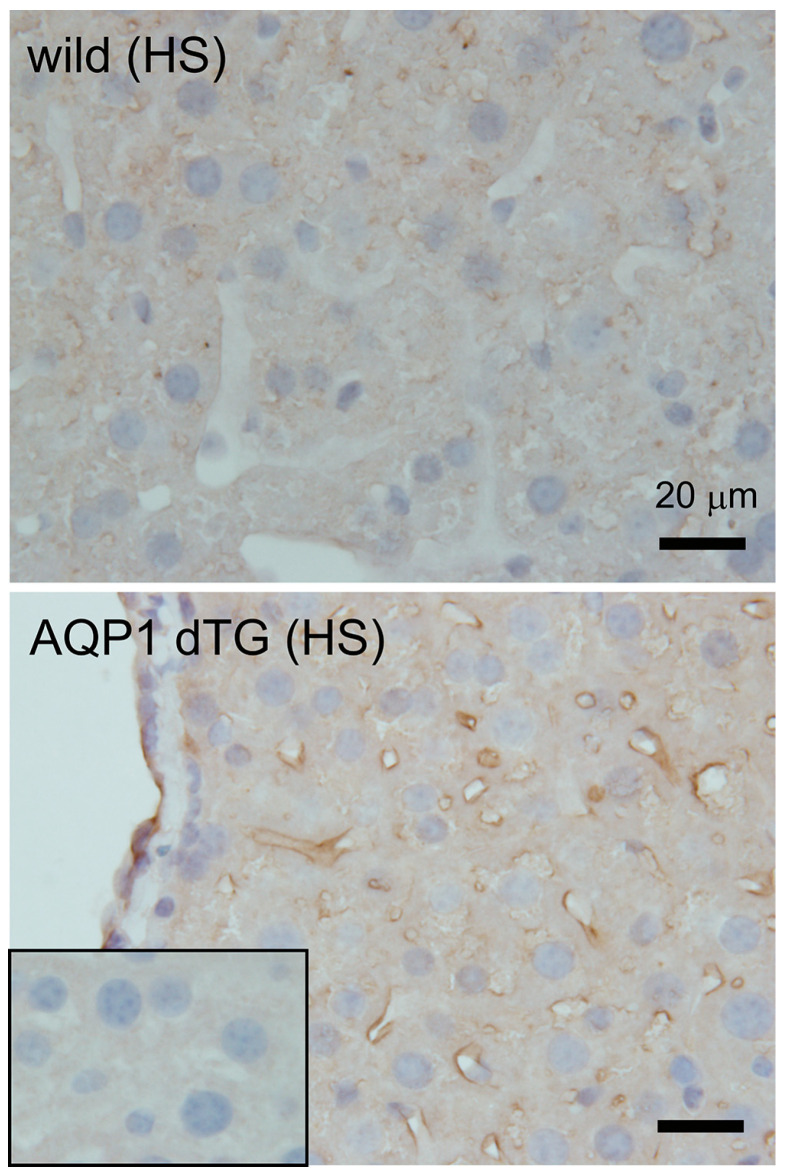
Immunostaining of 3-nitrotyrosine (3-NT) in the liver. A protein oxidative metabolite, 3-NT, was immunostained in the liver after 1 day of heat exposure. Minimal or no staining was observed in the 1st AB-free NC (inset of bottom) in the livers of Tie2-Cre/LNL-AQP1 dTG mice. Slight 3-NT-ir was detected in wild-type mice (**top**), and it was prominent in the vessels, including the sinusoids of the livers in Tie2-Cre/LNL-AQP1 dTG mice (**bottom**).

**Figure 8 biomedicines-12-02057-f008:**
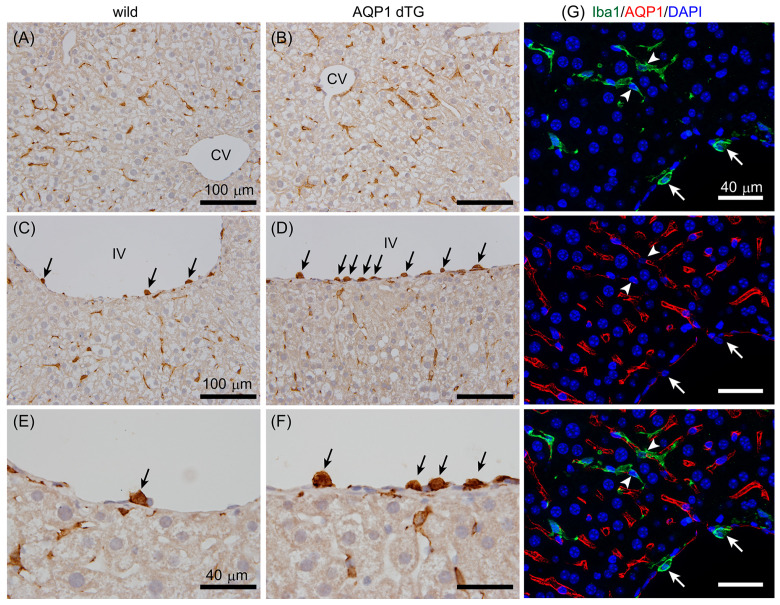
Iba1 immunostaining in the liver. A red-brown Iba1-ir was observed in the sinusoids, suggesting the presence of Kupffer cells, and it did not show a remarkable difference between wild-type (**A**) and Tie2-Cre/LNL-AQP1 dTG (**B**) mice. Iba1-ir (black arrows) was also observed in small- and medium-sized vessels, such as interlobular vessels (IVs), and was more prominent in Tie2-Cre/LNL-AQP1 dTG (**D**,**F**) mice than in wild-type mice (**C**,**E**). (**E**,**F**) show high-magnification images of Iba1-ir on IV. (**G**) Multiple staining for Iba1 (green) and AQP1 (red) was performed to determine whether Iba1^+^ cells expressed AQP1. Iba1+ cells were observed in Kupffer cells (arrowhead) and vascular adherent monocytes/macrophages (white arrows) but did not co-localize with AQP1-ir. Blue 4′,6-diamidino-2-phenylindole (DAPI) staining indicates nuclei. CV, central vein; IV, interlobular vessel; scale bars, 100 μm (**A**–**D**) and 40 μm (**E**–**G**).

**Figure 9 biomedicines-12-02057-f009:**
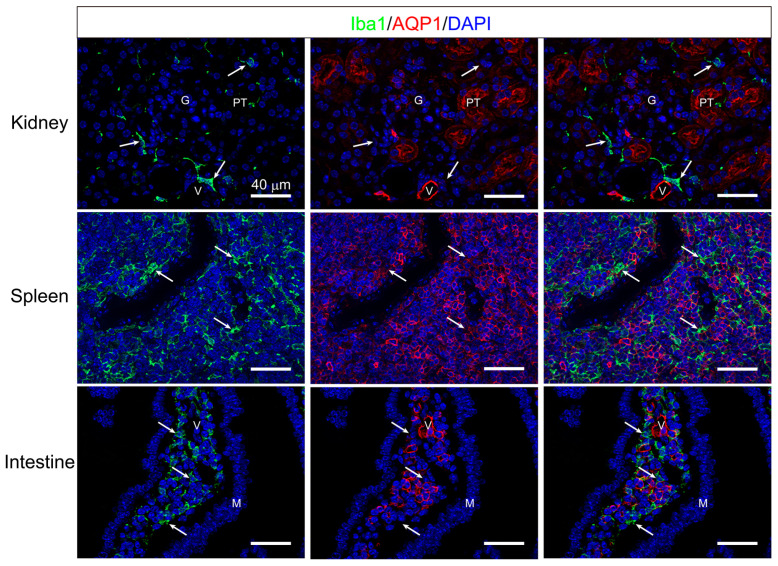
Multiple immunostaining of Iba1 and AQP1. Multiple staining of pan-monocyte/macrophage markers, Iba1 (green) and AQP1 (red), were performed in the kidney (upper), spleen (middle), and intestine (bottom), in addition to the liver (see Figure 8) to eliminate the expression of AQP1 in monocyte and macrophage lineage cells in Tie2-Cre/LNL-AQP1 dTG mice. The AQP1- ir was observed in the proximal tubules (PTs) and vasculature (V), including the glomeruli (G). However, Iba1-ir (arrow) was localized in the stroma and did not co-localize. In the spleen, the area of red pulp is recognized by many AQP1-ir molecules. While Iba1-ir (arrow) was also recognized in the red pulp, it did not co-localize. The AQP1-ir in the intestine was observed in the lamina propria but not in the mucous epithelia (M). Some were likely stromal cells, whereas others formed vasculature-like (V) structures. Although many Iba1-irs (arrow) were also recognized in the lamina propria, they did not co-localize. Blue DAPI staining indicates nuclei. Scale bar: 40 μm.

**Figure 10 biomedicines-12-02057-f010:**
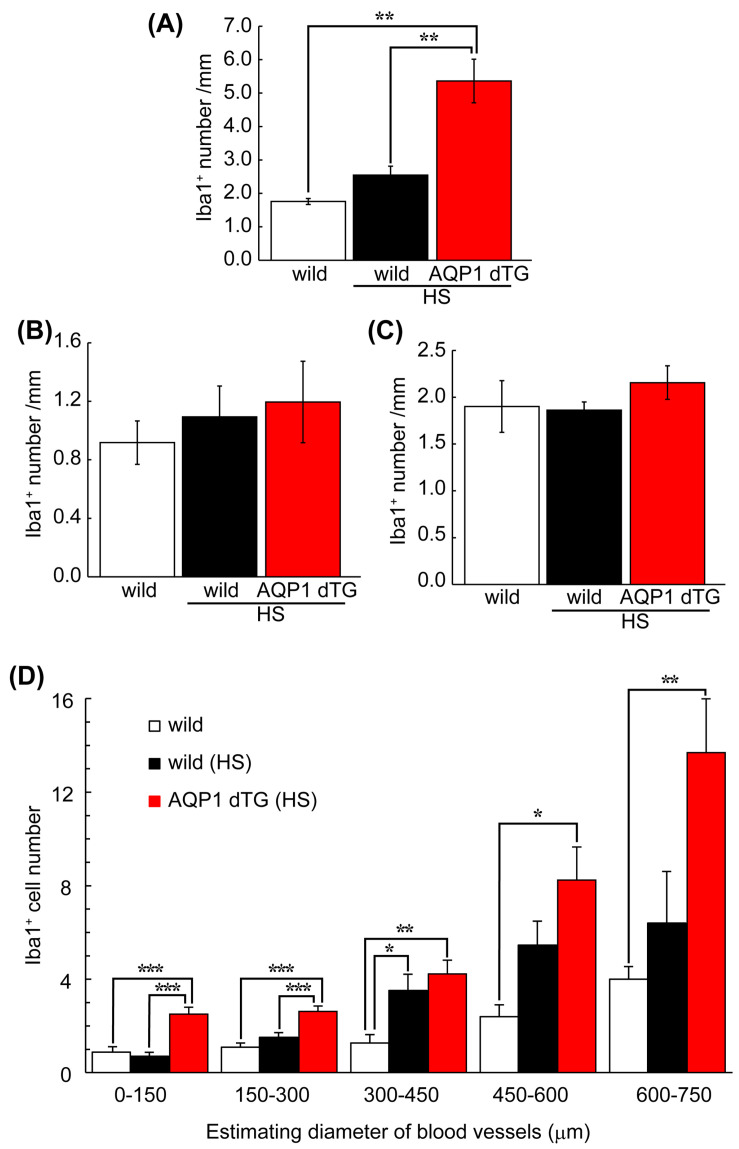
Comparison of Iba1-positive numbers on the blood vessels of the liver. Vasculature-adherent Iba1-positive cells (Iba1^+^) were counted in 30 vessels of the liver (**A**) and brain (**B**) and 20–24 vessels of the kidney (**C**) in each animal. The results were expressed as the mean number and length of the vessels (mm). (**A**) The number of Iba1^+^ cells was significantly higher in Tie2-Cre/LNL-AQP1 dTG mice (n = 5) than in wild-type mice, both with (n = 4) and without (n = 3) heat exposure. The numbers did not differ between the brain and kidneys. (**D**) The number of Iba1^+^ cells in the liver was used to estimate the diameter of each blood vessel, calculated from the circumferential length. While Iba1+ numbers in Tie2-Cre/LNL-AQP1 dTG mice were greater in all vessel sizes, the diameters of vessels measuring 0–150 and 150–300 μm differed significantly from those in wild-type mice after heat exposure. The data are expressed as mean ± SE. * *p* < 0.05, ** *p* < 0.01, *** *p <* 0.001 (Tukey’s post hoc test).

**Table 1 biomedicines-12-02057-t001:** List of primers.

Gene Symbol	Amplicon Size	Fwd	Rev	Ref Seq Numbers
*Aqp1*	124	AGGCTTCAATTACCCACTGGA	GTGAGCACCGCTGATGTGA	NM_007472
*Pecam*	136	TGGTTGTCATTGGAGTGGTC	TTCTCGCTGTTGGAGTTCAG	NM_008816
*Pecam*	812	GGTGACCTCCAATGACCCAG	GCCTTCCGTTCTTAGGGTCG	NM_008816
*Vwf*	125	CTTCTGTACGCCTCAGCTATG	GCCGTTGTAATTCCCACACAAG	NM_011708.4
*Eno2*	302	TGAGAATAAATCCTTGGAGCTGGT	GGTCATCGCCCACTATCTGG	NM_001302642
*Gfap*	419	CTAACGACTATCGCCGCCAA	CTGGTGAGCCTGTATTGGGAC	NM_001131020
*Mbp*	262	CAGAGTCCGACGAGCTTCAG	CAGCTTCTCTACGGCTCGG	NM_001025245
*Aif1*	144	ATCAACAAGCAATTCCTCGATGA	CAGCATTCGCTTCAAGGACATA	NM_019467
*Rps18*	166	AGTTCCAGCACATTTTGCGAG	TCATCCTCCGTGAGTTCTCCA	NM_011296

All primers were obtained from Eurofins Genomics (Tokyo, Japan).

## Data Availability

The datasets generated for this study are available on request from the corresponding author due the privacy.

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
