# Peer review of "Exacerbation of Hepatic Damage in Endothelial Aquaporin 1 Transgenic Mice after Experimental Heatstroke"

_biomedicines, 2024, doi:10.3390/biomedicines12092057_

Round 1

Reviewer 1 Report

Comments and Suggestions for Authors

Dear authors 

The paper is interesting although lacking concrete conclusions.

Is there any targetting therapeutic for AQP1? 

Heat exposure is important but it is not a cause of liver disease. Are there any interplay between AQP 1 and cirrhosis? Are there any influence regarding etiology of liver disease? Is AQP 1 associated with decompensation in cirrhosis?

Author Response

Dear Editor and Reviewer 1

Manuscript. Number.: biomedicines-3172427

Title: Exacerbation of hepatic damage in endothelial aquaporin 1 transgenic mice after experimental heatstroke

Journal: Biomedicines

We are extremely thankful to the Editor and the reviewers who helped to improve our manuscript. We have read all the comments and provided point-by-point responses below. The revised text in the manuscript is in yellow line.

Reviewer 1

  1. The paper is interesting although lacking concrete conclusions.

Thank you very much. Maybe you know, AQP1 is fist isolated AQP family from red blood cells, and expresses in kidney, red blood cells, choroid plexuses, endothelial cells (ECs, but not cerebral endothelial cells), and the others. We have reported that our heat stroke mice were impaired livers, kidneys, intestines and cerebellum of brain (Miyamoto et al. 2021, 2022). Therefore, we first expected to be influenced the effect of transgene in the brain because cerebral ECs are not expressed AQP1 in general and increased the expression in this transgenic mouse. However, contract of our expectation, the AQP1 Tg mice exhibited to a hepatic phenotype that is increases of Iba-1 positive macrophages and an oxidative stress after heat exposure. Therefore, we guess now that either increase of AQP1 in hepatic ECs acerates to hepatic damage or a signal of hepatic damage induced the expression or translocation of AQP1. We believe that our present results after heat stroke were really new, but we agree with you to need further investigation to clarify the role of endothelial AQP1.

  1. Is there any targeting therapeutic for AQP1? 

Thank you very much. That’s a point of our study. If we had been able to clarify the role of AQP1 in blood vessels more clearly, we might have been able to discuss more clearly.

So far, there are no therapeutic medicine to target for AQP1. However, it has been reported that the gene deficient mice of AQP1 decreased urinary concentration. The water homeostasis is one of key target for prevention of heat illnesses. Therefore, we can sure that AQP1 might contribute to modulate the heat illness. We are thinking if we can determine the role of AQP1 more clearly, we want to try inhibition or something other way.

  1. Heat exposure is important but it is not a cause of liver disease. Are there any interplay between AQP 1 and cirrhosis? Are there any influence regarding etiology of liver disease? Is AQP 1 associated with decompensation in cirrhosis?

Thank you for your supportive comments. We agree with you. There is no direct evidence with the relation between AQP1 and cirrhosis so far. However, as shown in our discussion (L582-585), sauna-associated death and fatal heat stroke have been shown increased AQP3 immunoreaction in the skin epidermis and Aqp4 gene expression in the brain. In present study, it is suggested the relation between AQP1 and heat-related illness. On the other hand, AQP1 expression increases in cirrhotic liver ECs following bile duct ligation. AQP1 gene-deficient (KO) mice with cirrhosis show decreased angiogenesis, fibrosis, and portal hypertension, which depend on osmotically sensitive microRNAs in the AQP1 pathway. These previous reports suggest that AQP1 involved in the hepatic diseases. However, our interpretation after these sentences was slightly mislead as your comments. We changed the expression to the others.

Reviewer 2 Report

Comments and Suggestions for Authors

The present study focused on a very interesting word problem about the effect of global warming on human health. Although global warming is expected to continue increasing and the understanding of the pathogenesis of heatstroke and as a consequence the developing of effective therapeutic strategies is an important challenge.

In their transgenic animal model, Aquaporin 1 (AQP1), key protein for water homeostasis, was chosen to be the target for the exacerbation of hepatic damage after experimental heatstroke.

In general, the manuscript is well written, and the experimental result supports the conclusion of the study. The article can be accepted in this journal and the author is invited to answer/explain the first comments round.

-Line 108: The confirmation of the dTG mice by the gene expression of Aqp1 in the brain because brain ECs generally do not express Aqp1is not sufficient due to endothelial cells expressing very low levels of AQP1based in different study. The author should explain more this point and can introduce more references.

-Line 179: Insert reference about the use of 3h of dehydration period.

-Line 185: what is the necessity of anesthesia with an overdose of sodium pentobarbital?

 - Line 262: correct the sentence "and in" in place of "andin"

- The morphological alterations of hepatic tissue in the wild-type and Tie2-Cre/LNL-AQP1 dTG mice is well explained in the present study. Although, the author doesn’t pay any attention to the other tissues (kidney for example). This can improve the manuscript even without any damage. If there are additional result (hematoxylin-eosin staining) for other tissues, it will be very interesting.

Author Response

Dear Editor and Reviewer 2

Manuscript. Number.: biomedicines-3172427

Title: Exacerbation of hepatic damage in endothelial aquaporin 1 transgenic mice after experimental heatstroke

Journal: Biomedicines

We are extremely thankful to the Editor and the reviewers who helped to improve our manuscript. We have read all the comments and provided point-by-point responses below. The revised text in the manuscript is in yellow line.

Reviewer 2

  1. The present study focused on a very interesting word problem about the effect of global warming on human health. Although global warming is expected to continue increasing and the understanding of the pathogenesis of heatstroke and as a consequence the developing of effective therapeutic strategies is an important challenge. In their transgenic animal model, Aquaporin 1 (AQP1), key protein for water homeostasis, was chosen to be the target for the exacerbation of hepatic damage after experimental heatstroke. In general, the manuscript is well written, and the experimental result supports the conclusion of the study. The article can be accepted in this journal and the author is invited to answer/explain the first comments round.

Thank you very much for your worm and positive comments.

  1. -Line 108: The confirmation of the dTG mice by the gene expression of Aqp1 in the brain because brain ECs generally do not express Aqp1 is not sufficient dueto endothelial cells expressing very low levels of AQP1 based in different study. The author should explain more this point and can introduce more references.

Thank you very much for your comments. Most of review articles we checked described not to express AQP1 in brain endothelial cells of capillaries. But as you mentioned, there is a report that primary brain capillary endothelial cells of rat express low level of AQP1 (J Neurochem 2005). We added the reports and changed the description in these parts.

  1. -Line 179: Insert reference about the use of 3h of dehydration period.

Thank you very much. We have missed to refer our study and inserted the references.

  1. -Line 185: what is the necessity of anesthesia with an overdose of sodium pentobarbital?

Now a days, to terminate animals, overdose anesthesia, approximately twice of suitable doses is recommended in the point of view for animal care.

  1. Line 262: correct the sentence "and in" in place of "andin"

Thank you. we did it.

  1. The morphological alterations of hepatic tissue in the wild-type and Tie2-Cre/LNL-AQP1 dTG mice is well explained in the present study. Although, the author doesn’t pay any attention to the other tissues (kidney for example). This can improve the manuscript even without any damage. If there are additional result (hematoxylin-eosin staining) for other tissues, it will be very interesting.

Thank you for your comments. In present study, we could not see any serum biochemical changes which indicate renal damage such as Cre and BUN. Therefore, we did not focus on the morphological comparison among them. However, in case of severe damages in kidney, the animals were exhibited abnormality mainly in proximal tubules. The tubules showed swelling and degeneration of tubular epithelial cells and urinary casts. Please see out previous reports (Miyamoto 2021).

Round 2

Reviewer 1 Report

Comments and Suggestions for Authors

After the corrections, I am satisfied.